# Combined toll-like receptor 3/7/9 deficiency on host cells results in T-cell-dependent control of tumour growth

Johanna C. Klein[1], Katrin Moses[1], Gennadiy Zelinskyy[2], Simon Sody[1], Jan Buer[3], Stephan Lang[1,4], Iris Helfrich[4,5], Ulf Dittmer[2], Carsten J. Kirschning[3,*] & Sven Brandau[1,4,*]

Toll-like receptors (TLRs) are located either on the cell surface or intracellularly in endosomes and their activation normally contributes to the induction of protective immune responses. However, in cancer their activation by endogenous ligands can modulate tumour progression. It is currently unknown how endosomal TLRs regulate endogenous anti-tumour immunity. Here we show that TLR3, 7 and 9 deficiencies on host cells, after initial tumour growth, result in complete tumour regression and induction of anti-tumour immunity. Tumour regression requires the combined absence of all three receptors, is dependent on both CD4 and CD8 T cells and protects the mice from subsequent tumour challenge. While tumours in control mice are infiltrated by higher numbers of regulatory T cells, tumour regression in TLR-deficient mice is paralleled by altered vascular structure and strongly induced influx of cytotoxic and cytokine-producing effector T cells. Thus, endosomal TLRs may represent a molecular link between the inflamed tumour cell phenotype, anti-tumour immunity and the regulation of T-cell activation.

[1] Research Division of the Department of Otorhinolaryngology, University Hospital Essen, University Duisburg-Essen, Hufelandstrasse 55, D-45122 Essen, Germany. [2] Institute of Virology, University Hospital Essen, University Duisburg-Essen, D-45122 Essen, Germany. [3] Institute of Medical Microbiology, University Hospital Essen, University Duisberg-Essen, D-45122 Essen, Germany. [4] German Cancer Consortium (DKTK), D-45122 Essen, Germany. [5] Skin Cancer Unit of the Department of Dermatology, West German Cancer Center, University Hospital Essen, University Duisberg-Essen, D-45122 Essen, Germany. * These authors jointly supervised this work. Correspondence and requests for materials should be addressed to S.B. (email: sven.brandau@uk-essen.de).

Toll-like receptors (TLRs) are a conserved family of receptors, well recognized for their ability to respond to pathogenic structures, also known as pathogen-associated molecular patterns[1]. TLRs are located either on the cell surface or intracellularly in endosomes. Although surface TLRs such as TLR2 and TLR4 primarily recognize bacterial proteins, endosomal TLRs primarily detect viral and bacterial nucleic acids[2]. Triggering of TLRs initiates a complex intracellular signalling cascade in activated cells. Among others, MyD88 and TRIF are major adaptor molecules in this cascade. In the immune system, TLR ligation results in the activation of myeloid cells and subsequent induction of anti-pathogenic immunity[3]. In this context, the activation of myeloid immune cells via TLRs represents a link between innate and adaptive immunity[4].

This immunostimulatory potential of TLR ligation has been used to develop cancer immunotherapies based on synthetic or natural TLR ligands. Examples for both therapeutic ligands of membrane bound as well as endosomal TLRs exist. Compounds targeting endosomal TLRs and mimicking viral and bacterial RNA and DNA have attracted a somewhat broader interest. Examples include the use of imiquimod and CpG, ligands for the endosomal TLRs 7 and 9. Imiquimod is a short synthetic RNA and is clinically used to treat actinic keratosis, external genital warts and superficial basal cell carcinoma[5]. CpG, through binding to TLR9, has strong adjuvant activity and has been applied in numerous clinical trials in the treatment of allergy, cancer and infectious diseases[6].

In the context of these therapeutic applications the expression of TLRs on tumour cells and the consequences of TLR activation on cancer cells have received increasing attention. Studies in this area quickly revealed the dichotomous nature of tumour cell stimulation with TLR ligands. On the one hand, triggering of TLRs has been reported to induce cell death in tumour cells[7,8]. This cell death may be anti-tumoural in two ways: First, as a direct consequence, the number of tumour cells is reduced. Second, via a process termed 'immunogenic cell death', additional activation of anti-tumour immunity may occur[9,10]. However, some evidence suggests that this type of immunogenic cell death may rather be associated with the effects of TLR ligands on RIG-I-like helicases[11], another class of pattern recognition receptors responsive to synthetic and pathogenic nucleic acids. Conversely, under certain conditions, TLR ligands may also elicit anti-apoptotic effects[12], which can even be associated with additional escape from cytolysis by immune cells[13].

Although the effects on cancer (and epithelial) cells are fairly well understood, the consequences of TLR signalling in fibroblasts and mesenchymal cells remain poorly described. Some studies suggest tumour-promoting effects of TLR expression on cancer-associated fibroblasts. For example, TLR4 expression by stromal fibroblasts in colorectal cancer was associated with poor prognosis[14] and overexpression of TLR3 in fibroblasts results in upregulation of the oncoprotein c-Myc[15]. Other studies showed that high TLR9 messenger RNA expression on fibroblast-like cells in breast or oesophageal squamous cell carcinoma was associated with reduced metastasis and invasion[16,17].

In addition to foreign pathogenic structures, the TLRs also recognize non-foreign structures commonly referred to as damage-associated molecular pattern molecules (DAMPs). DAMPs are self-proteins of the host, which are released during pathogenic conditions such as chronic inflammation, autoimmune diseases, sepsis and cancer[18].

In cancer, DAMPs are released as a consequence of continued cancer-related inflammation, tissue destruction and cell death[19]. The release of the nuclear DNA-binding molecule high-mobility group box 1 (HMGB1) by dying tumour cells activates TLR2 and TLR4 on DCs, resulting in an efficient processing and cross-presentation of tumour antigens and tumour regression[20,21]. Similarly, members of the S100 family are released during cellular stress and cancer. In immune cells, this leads to TLR4-dependent release of pro-inflammatory cytokines and receptor for advanced glycation endproducts (RAGE)-dependent migration[22]. In addition, histones, if translocated from the nucleus to the extracellular space, function as DAMPs[23] and increased levels of circulating histones were found, for example, in serum of pancreatic cancer patients[24]. Likewise, nucleic acid fragments released from necrotic cancer cells or adjacent injured epithelial cells can act as DAMPs[25,26].

Thus, despite some uncertainties, the continued triggering of TLRs by endogenous ligands is mainly associated with tumour-promoting effects[27]. These effects are mostly linked to TLR2 and TLR4. The role for the endosomal TLRs such as TLR3, 7 and 9 is less clear and is investigated in this study. We use syngeneic wild-type (WT) transplantable tumour cells and TLR-deficient recipient mice, to specifically address the role of TLR expression on host cells. During the first 7–10 days, tumour growth is virtually identical in WT and Tlr3/7/9-deficient mice. Unexpectedly, post day 10 a clear tumour rejection is observed in Tlr triple knockout (KO) mice. This phase is associated with a dramatic local inflammatory reaction and substantial changes in the tumour microenvironment, ultimately enabling an unexpected T-cell-dependent complete tumour rejection. In sum, our data suggest that TLR signalling may provide a molecular link between tumour-associated inflammation and anti-tumour immunity.

## Results

**Tumour rejection in the absence of TLR signalling**. To investigate the role of the endosomal TLRs 3, 7 and 9 on tumour growth, a murine head and neck cancer cell line (murine oropharyngeal carcinoma cell, MOPC) was injected into single KO, double KO and Tlr3/7/9 triple KO mice (Fig. 1a and Supplementary Fig. 1). In the majority of TLR-deficient and WT control mice, a palpable tumour developed within 1 week. Data from single KO mice were comparable to WT mice and thus were not presented in the figures. Unexpectedly, at later points in time, tumours declined in size in Tlr3/7/9 triple KO, but not single or double KO mice, and fully disappeared between day 14 and day 30 (Fig. 1a and Supplementary Fig. 1). In contrast, only one out of six $Tlr3/9^{-/-}$ mice could reject the tumour and only two single KO mice did not develop a tumour after 3–4 weeks. Histologic analysis of tumour sections, taken from WT and $Tlr3/7/9^{-/-}$ mice at different time points during the initial growth and the later rejection phase, confirmed tumour regression (Fig. 1b). To exclude cell line-specific effects, two additional syngeneic tumour cell lines (TC1, MB49) were injected into Tlr triple KO mice. Cell numbers for the inoculum were determined according to titration experiments performed in WT mice. Again, all TLR3/7/9-deficient mice and WT mice developed tumours during the initial 1–2 weeks post injection (depending on the cell line used), but only Tlr triple KO mice rejected tumours within 40 days (Supplementary Fig. 2).

To test whether tumour regression is caused by tumour cell death, we performed TdT-mediated dUTP nick end labelling (TUNEL) staining on tissue sections. Although only little cell death could be detected in both WT and $Tlr^{-/-}$ mice during the first week of tumour growth, a dramatic increase in TUNEL-positive cells was observed in TLR-deficient mice at day 10 and day 14 (Fig. 1c,d), suggesting substantial tumour cell death during regression.

Tumour growth may be influenced by the local tissue micro-environment. To explore whether this phenomenon is limited to

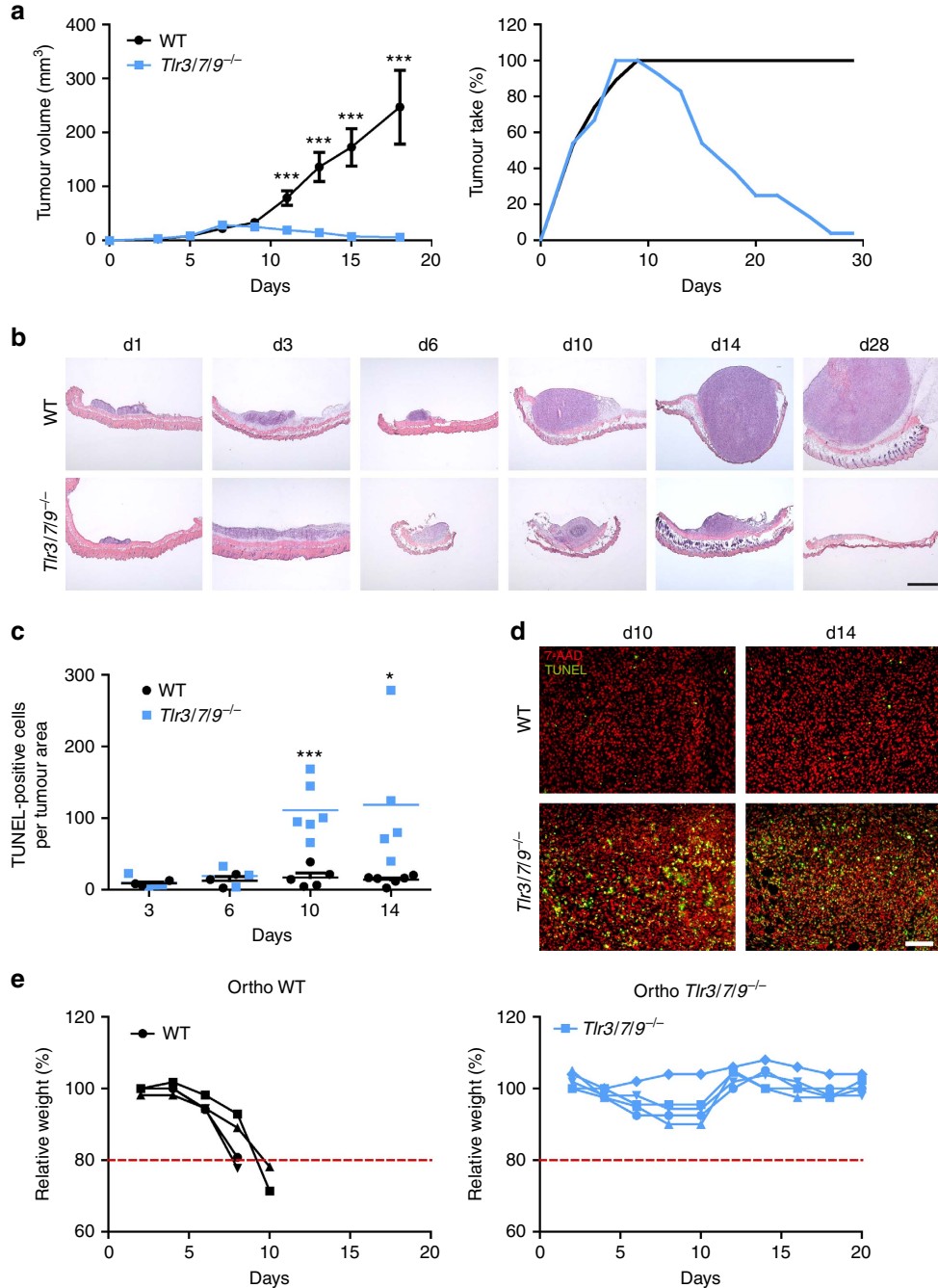

**Figure 1 | *Tlr3/7/9*$^{-/-}$ mice reject MOPC tumours within 3 weeks through induction of cell death.** (**a**) MOPC tumour cells ($0.5 \times 10^6$) were s.c. injected into the right flank of 8- to 16-week-old C57BL/6 WT ($n = 19$) and *Tlr3/7/9*$^{-/-}$ mice ($n = 24$). Mean of tumour growth ± s.e.m. and tumour take (percentage of mice with a palpable tumour) are shown. Three stars (***) indicate statistical difference $P \leq 0.0005$ (*t*-test). (**b**) Frozen sections of s.c. tumours, dissected at different time points after tumour cell injection (day 1 to day 28), were stained by haematoxylin–eosin staining. One representative example for each time point is shown. Scale bar, 1,000 µm. (**c**) Quantification of dead cells in MOPC tumours was determined by calculating the area of TUNEL-positive cells in relation to the entire tumour area at day 3, day 6, day 10 and day 14 after tumour cell injection ($n = 3$–6). Single values and mean (horizontal line) are shown. Results were considered significant at *$P \leq 0.05$ and *** $P \leq 0.0005$ (*t*-test). (**d**) Representative examples of the TUNEL staining at day 10 and day 14 of tumours of WT and *Tlr3/7/9*$^{-/-}$ mice. Green: TUNEL, red: 7-AAD nuclear staining. Scale bar, 100 µm. (**e**) Tumour growth of $0.5 \times 10^6$ MOPC cells after orthotopic (ortho) injection into the floor of the mouth in 8- to 12-week-old C57BL/6 WT ($n = 4$) and in *Tlr3/7/9*$^{-/-}$ mice ($n = 5$). Relative weight of the mice is shown. Mice were killed if they lost 20 % of their initial weight, indicated by the dotted red line.

the subcutaneous (s.c.) injection site, we next extended our analysis to an orthotopic model recently developed by our group[28]. In this model, the tumour cells are injected into the floor of the mouth and the weight of the mice is recorded as a measure of tumour growth. WT and *Tlr*$^{-/-}$ mice started to lose weight during the initial growth phase as a consequence of impaired food uptake. Around day 10, analogous to the s.c. model, *Tlr*$^{-/-}$ mice started to recover to normal body weight. In contrast, WT mice showed progressive tumour growth and were killed according to animal welfare regulations (Fig. 1e). These data indicate that the

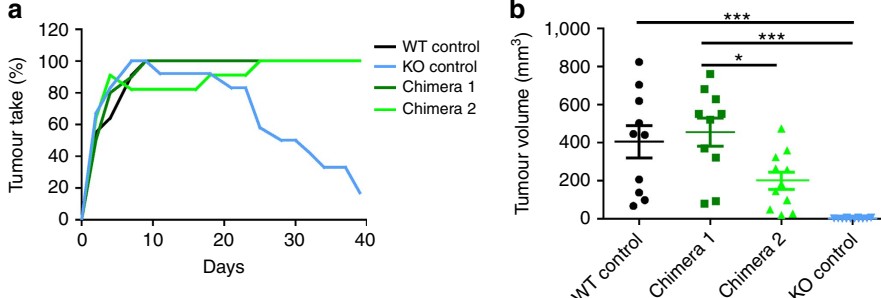

**Figure 2 | Haematopoietic and non-haematopoietic stromal components are required for tumour rejection.** BM chimeras (8–14 weeks old) were generated via irradiation of recipient mice and subsequent BM transfer from donor mice 1 day later. WT mice received BM from $Tlr3/7/9^{-/-}$ mice to generate chimera 1 (dark green). To generate chimera 2 (light green), $Tlr3/7/9^{-/-}$ mice received WT BM. As a control, irradiated WT mice received WT BM (black) and $Tlr3/7/9^{-/-}$ mice received TLR-deficient BM (blue) ($n = 10$–12). Eight weeks later, MOPC cells were injected s.c. and tumour take (number of mice with palpable tumour) was recorded (**a**). Tumour volume 30 days after tumour injection of individual mice and mean (horizontal line) ± s.e.m. are shown; $P$-value (analysis of variance) is indicated (**b**). Results were considered significant at *$P \leq 0.05$ and *** $P \leq 0.0005$.

mechanism of tumour rejection is not limited to the s.c. site, but rather maintained at the orthotopic site.

**Immune and non-immune cells are involved in tumour rejection.** The stromal tumour microenvironment consists of immune cells and non-immune stromal cells. Although immune cells are of haematopoietic origin, the remaining stromal cells are largely non-haematopoietic and contain vessels and mesenchymal fibroblastoid cells. We performed classical bone marrow chimera experiments, to test whether tumour rejection is mediated via TLR deficiency on haematopoietic or non-haematopoietic cells. As depicted in Fig. 2a, tumour rejection only occurred in irradiated $Tlr^{-/-}$ mice, which received $Tlr^{-/-}$ bone marrow (BM; designated KO control). Both chimera types (WT BM into KO host and KO BM into WT host) were permissive for tumour outgrowth, suggesting that TLR deficiency on both haematopoietic and non-haematopoietic stromal cells is required for complete tumour rejection. When we quantified tumour growth by caliper measurements on day 30 of the experiment (Fig. 2b), we found an expected great variance in tumour size between individual animals. No or very small tumours could be measured in KO control mice, whereas tumours were clearly measurable in WT control and in both types of chimera. These data suggest that cooperative activity of both haematopoietic and non-haematopoietic, radioresistant TLR-deficient stromal cells is required for tumour rejection.

Blood vessels are a major component of the non-haematopoietic stromal tumour microenvironment. When we analysed the vessel composition by CD31 immunohistochemistry between day 6 and 14, we found substantial differences between WT and $Tlr^{-/-}$ mice. At day 10 and 14, gross microscopic inspection (Fig. 3a) already suggested a considerable increase in vessel diameter in $Tlr^{-/-}$ mice. This increase was then confirmed and quantified using digital image analysis showing a twofold increase in vessel diameter in the $Tlr^{-/-}$ group compared with WT mice (Fig. 3b). During the regression phase (days 10 and 14), the enhanced vessel diameter was accompanied by a decrease in the number of vessels per microscopic field (Fig. 3c). However, on day 6, before the regression phase, no differences in vessel diameter and density were observed. To substantiate these findings from two-dimensional immunohistochemical analysis, we used light sheet microscopy and three-dimensional (3D) analysis (Fig. 3d and Supplementary Movies 1 and 2). Tumours in WT mice showed a typical pro-angiogenic dense microvascular network, which was distributed through the entire tumour.

Interestingly, during the rejection phase (day 10), tumours in $Tlr^{-/-}$ mice were more heterogeneously vascularized and large tumour areas appeared to be devoid of a typical tumour-associated vascular network (Fig. 3d, lower panels). To examine whether endothelial cell death could be involved in this phenomenon, we combined CD31 staining with caspase staining. Most endothelial cells were negative for caspase in tumours of WT and $Tlr3/7/9^{-/-}$ mice, making substantial endothelial caspase-mediated cell death during tumour rejection unlikely (Fig. 3e). We next analysed the vascular maturation level in both tumour types by using the established pericyte markers desmin and α-smooth muscle actin[29]. In accordance with the enhanced vascular diameter found in $Tlr^{-/-}$ mice, at day 14 intratumoural microvessels of $Tlr^{-/-}$ mice displayed a mature vascular network with strong recruitment of desmin-positive mural cells and indicative of vessel normalization. In contrast, total loss or partial lack of pericyte coverage was observed in tumours of WT mice (Fig. 3f). Quantification of vessel maturation indicated no difference at day 10 (WT 52 % Desmin$^+$ vessels; $Tlr^{-/-}$ 59% Desmin$^+$ vessels), but a clear difference became apparent at day 14 (WT 36 % versus $Tlr^{-/-}$ 91 %) (Fig. 3g). These data establish that the tumour regression in $Tlr^{-/-}$ mice is accompanied by vessel normalization, whereas tumour growth in WT mice is associated with a typical tumour-associated pro-angiogenic vessel phenotype.

**Inflamed tumour cell phenotype in TLR-deficient mice.** Based on these findings, we speculated that differences in the microvascular endothelium could be further associated with a differential infiltration of the tumours by immune cells. We found that tumours in WT mice were moderately infiltrated by immune cells (Fig. 4a,b). $Tlr^{-/-}$ mice showed a similar moderate infiltration until day 6 post injection. Post day 6, however, there was a dramatic increase in the number of tumour-associated leukocytes in $Tlr^{-/-}$ mice. This strongly induced immune infiltrate was composed of both myeloid (CD11b, Gr-1) and lymphoid (CD4, CD8) cells. Interestingly, this change in tumour infiltration was not accompanied by changes in the composition of peripheral blood immune cell subsets (Supplementary Fig. 3a). To test whether the inflammatory growth of tumours in $Tlr^{-/-}$ mice was indeed caused by enhanced and active migration of immune cells into the tumour tissue, we performed adoptive transfer experiments. To this end, tumour-bearing WT and $Tlr^{-/-}$ recipient mice were injected with fluorescently labelled splenocytes derived from tumour-bearing donor mice (both WT

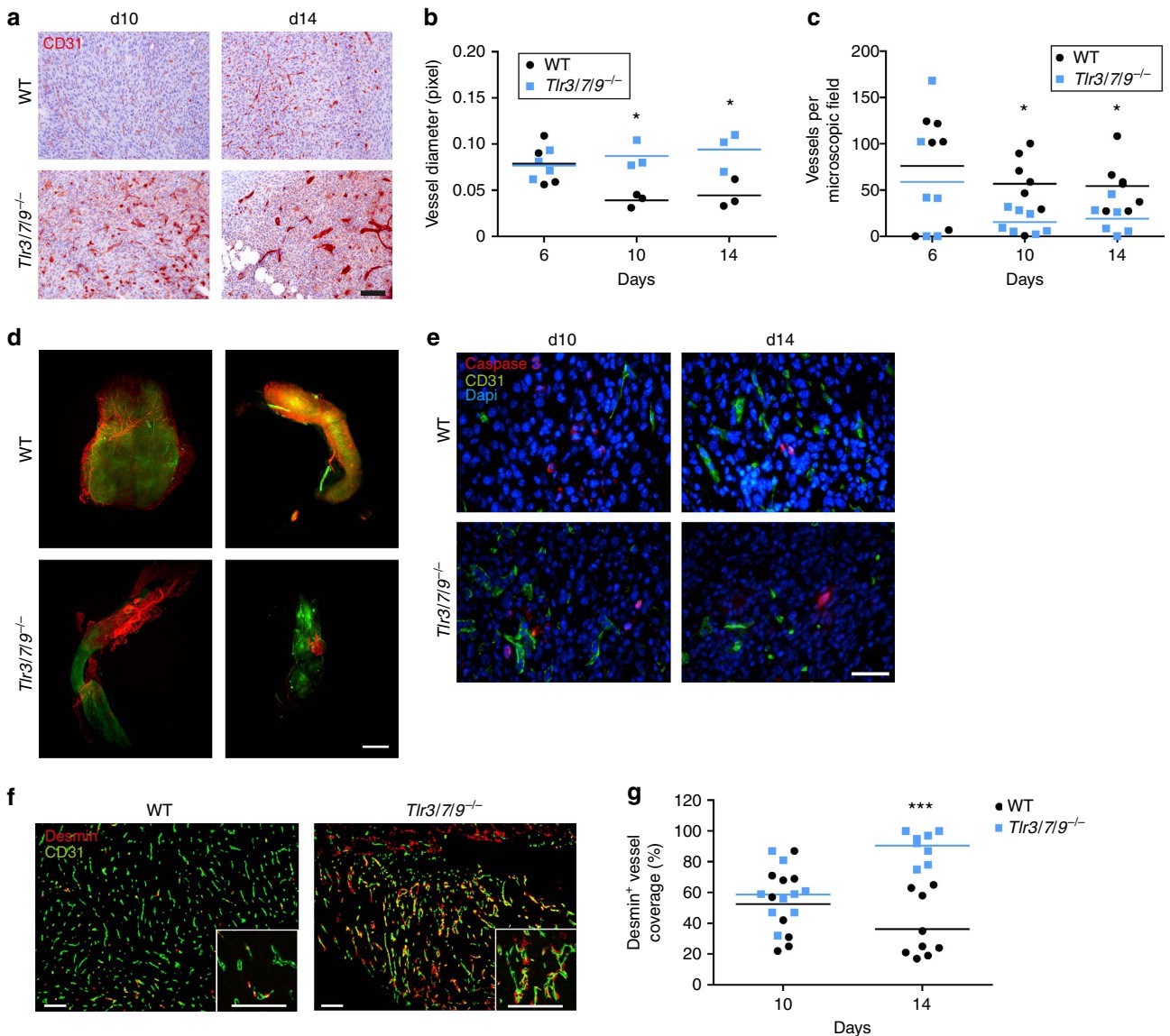

**Figure 3 | Increased vessel diameter and pericyte coverage in $Tlr^{-/-}$ mice.** (a–c) Frozen sections of s.c. tumours dissected at day 6, day 10 or 14 after injection were stained with αCD31 antibody, to assess the vascular system of tumours of 8- to 16-week-old C57BL/6 WT and $Tlr3/7/9^{-/-}$ mice. (a) Representative examples of tumours at day 10 und day 14. Scale bar, 100 μm. (b) For analysis of vessel diameter, vessels were assessed manually in ×200 magnified images and diameter was quantified using ImageJ software ($n=3$). P-value (t-test) is indicated (*$P\le0.05$). Single values and mean (horizontal line) are shown. (c) Quantification of vessel number was performed by manual observer assisted counting of four ×200 magnified randomly selected images per sample ($n=6$-7). Single values and mean (horizontal line) are shown. Results were considered significant at *$P\le0.05$ (t-test). (d) Twelve- to 16-week-old C57BL/6 WT and $Tlr3/7/9^{-/-}$ mice were perfused and s.c. tumours were surgically removed 10 days after injection of MOPC$^{-eGFP}$ tumour cells (green). Before tumour removal, vessels were stained in vivo by injection of Alexa Fluor 647-coupled αCD31 antibody (red). Tumour tissue was cleared and analysed by light sheet microscopy. Three-dimensional images were created using ImageJ and Imaris software. Shown are two representative examples of tumours of WT and TLR-deficient mice. Scale bar, 600 μm. (e) Three-colour immunofluorescence of explanted tumours. A representative example of CD31 (vessels) and caspase 3 (cell death) of d10 and d14 tumour sections from 8- to 16-week-old C57BL/6 WT and $Tlr3/7/9^{-/-}$ mice is shown. Dapi (4,6-diamidino-2-phenylindole; blue) was used to visualize nuclei. Scale bar, 50 μm. (f) Pericyte staining of vessels. A representative example of CD31 (vessels) and desmin (pericytes) of d14 tumour sections from 8- to 16-week-old C57BL/6 WT and $Tlr3/7/9^{-/-}$ mice is shown. Dapi (blue) was used to visualize nuclei. Scale bars, 100 μm. (g) Desmin-positive vessels from $n=3$ mice at day 10 and day 14 were quantified by manual counting of up to five fields per tumour, including 100 vessels in total, using the mean of three tumour section per tumour (top, middle and base) of concordant distance. Single values and mean (horizontal line) are shown. Three stars (***) indicate statistical difference $P\le0.0005$ (t-test).

and $Tlr^{-/-}$). One day later, tumours were explanted, cleared and subjected to 3D light sheet microscopy. As shown in Supplementary Fig. 3b, a substantial number of tumour-infiltrating splenocytes was only detectable in $Tlr^{-/-}$ mice, which received adoptively transferred cells from TLR-deficient mice. These data, in agreement with the BM chimera experiment

(Fig. 2), suggest that cooperative mechanisms between non-haematopoietic stroma and immune cells are responsible for the inflammatory immune cell recruitment in TLR-deficient mice.

In an attempt to better understand this inflammatory activity in tumours of $Tlr^{-/-}$ mice, we performed a pilot experiment (four mice, day 6) and analysed gene expression in tumour tissue

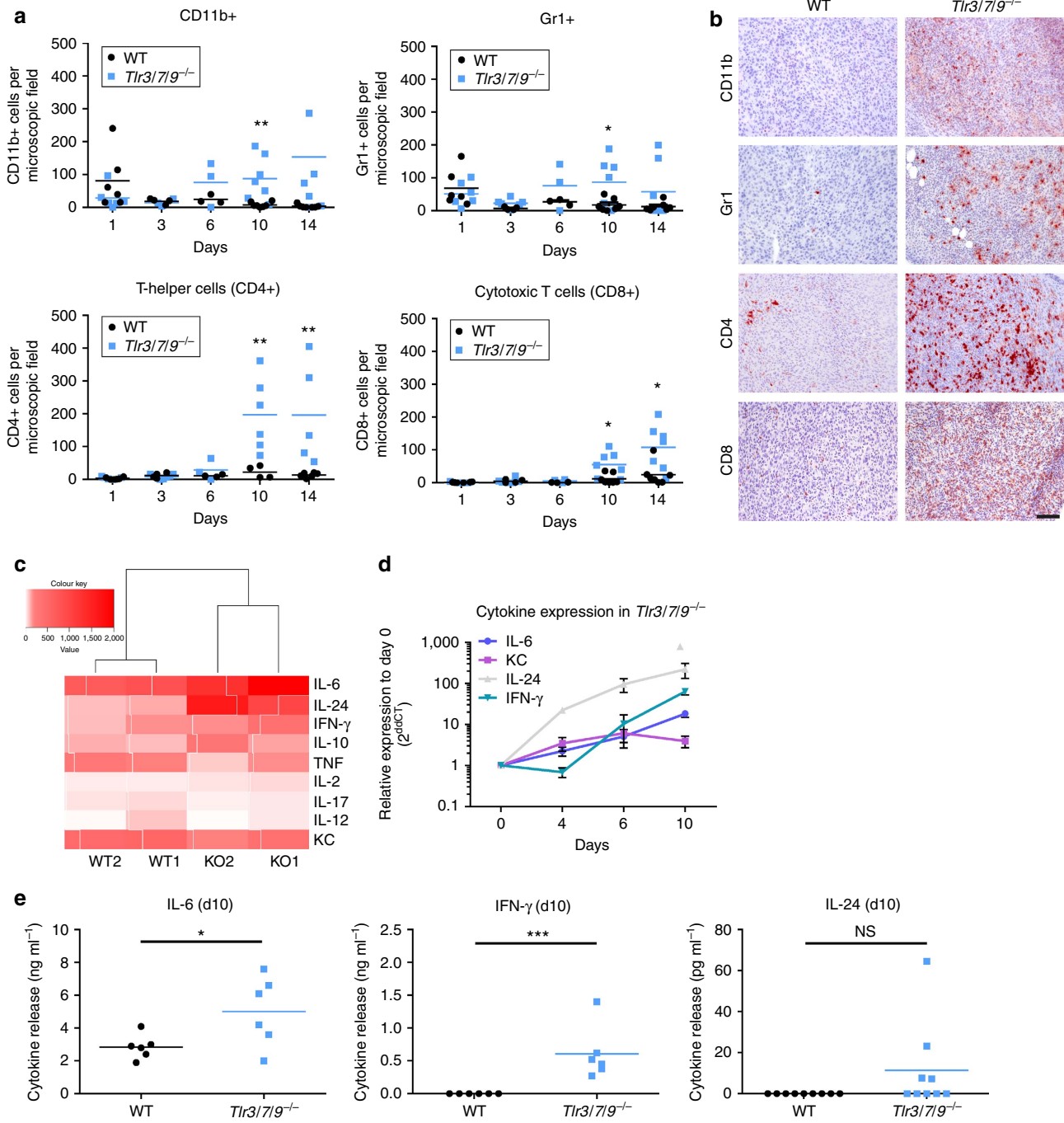

**Figure 4 | Immune cell recruitment and inflammatory tumour growth in *Tlr⁻/⁻* mice.** (**a**) Immunohistochemical staining of frozen tumour sections of 8- to 16-week-old C57BL/6 WT and *Tlr3/7/9⁻/⁻* was performed to assess the frequency of tumour-infiltrating immune cells at different time points (day 1, day 3, day 6, day 10 and day 14). Quantification of the number of positive cells was performed by observer-assisted counting on four randomly selected × 200 magnified images per sample (*n* = 3–7). Single values and mean (horizontal line) are shown. (**b**) Representative examples of immunohistochemical staining of tumours dissected 14 days after injection. Scale bar, 100 μm. (**c**) Hierarchical clustering of gene expression: 2 × 10⁶ MOPC tumour cells were injected s.c. into 8- to 16-week-old C57BL/6 WT and *Tlr3/7/9⁻/⁻* mice, and tumours were dissected 6 days later. RNA was isolated and a microarray analysis was performed using an Affimetrix Chip (*n* = 2). Signal intensity of different cytokines is shown as hierarchical clustering. Red indicates upregulation of gene expression; white indicates no gene regulation. (**d**) Gene expression of selected cytokines was analysed in a time course by quantitative PCR: day 0 (skin only), day 4, day 6 and day 10 (*n* = 3 mice each). Mean ± s.e.m. of relative expression to day 0 is shown for tumours of *Tlr3/7/9⁻/⁻* mice. (**e**) Tumours were explanted at day 10 after s.c. tumour cell injection into 10-week-old C57BL/6 WT and *Tlr3/7/9⁻/⁻*, and cultured for 2 days in medium (20 mg ml⁻¹). Concentration of IL-6, IFN-γ (*n* = 6) and IL-24 (*n* = 9) in the SNs was analysed. Single values and mean (horizontal line) are shown. Results were considered significant at *$P \leq 0.05$, **$P \leq 0.005$ and ***$P \leq 0.0005$ (*t*-test).

using RNA microarrays. Evaluation of cytokine gene expression suggested an induction of interleukin (IL)-6, IL-24 and interferon (IFN)-γ in KO over WT animals (Fig. 4c). These candidate genes were then further investigated in a time course and on the protein level (Fig. 4d,e). In agreement with the previous findings, induction of IL-6 and IL-24 started between day 4 and day 6,

and was followed by induction of IFN-γ. Release of IFN-γ and IL-6 was also confirmed on the protein level using a model of *ex vivo* cultured tumour explants (Fig. 4e). Induction of IL-24 seemed to be restricted to the RNA level and was not confirmed by enzyme-linked immunosorbent assay (ELISA) analysis of tumour explant supernatants (SNs).

**T-cell-dependent tumour rejection in TLR-deficient mice.** In the next series of experiments, we explored the activation state, phenotype and functionality of T cells in WT versus $Tlr^{-/-}$ mice. Proliferative activity of both CD4 and CD8 T cells from tumour-draining lymph nodes was higher in TLR-deficient mice compared with WT controls (Fig. 5a). In tumours, >40% of CD4 cells in TLR-deficient mice displayed an activated phenotype, defined as CD62L$^-$/CD154$^+$ (Fig. 5b). Similarly, the percentage of cytotoxic CD8 T cells (positive for granzyme B, CD107a or perforin) was strongly induced in TLR-deficient mice over WT mice (Fig. 5b and Supplementary Fig. 3d). In line with these findings, a substantial number of CD4 and CD8 cells from tumours of $TLR^{-/-}$, but not from WT mice, also secreted IFN-γ and tumour necrosis factor (TNF)-α in response to short-term *ex vivo* stimulation (Fig. 5c). In addition to activation markers and cytotoxic molecules, we also tested the expression of exhaustion markers and the frequency of intratumoural regulatory T cells. Although potential markers of exhaustion such as PD-1, LAG3 and Tim3 were not differentially expressed between both mouse strains, the percentage of FoxP3+ regulatory T cells was increased in WT mice over KO mice at day 10 (Supplementary Fig. 3c).

In conjunction, these data suggest increased proliferative activity in tumour-draining lymph nodes and increased effector T-cell activity in the tumour microenvironment of TLR-deficient mice. Interestingly, these differences were observed on day 10, but not on day 6, and thus coincide with the macroscopic rejection of the tumour.

The strong influx of T cells together with the major induction of IFN-γ and cytotoxic molecules in $Tlr^{-/-}$ mice prompted us to test the relevance of CD4 and CD8 cells for tumour rejection in depletion experiments. We also considered that potential anti-tumour T cells might be under the control of CD11b$^+$/Gr-1$^+$ regulatory myeloid cells and included Gr-1 depletion into our experimental design. In WT mice, tumour growth in isotype control-treated, CD4-depleted and CD8-depleted mice did not differ dramatically. However, slightly and statistically significant (*$P \le 0.05$ and ** $P \le 0.005$, *t*-test), accelerated tumour growth was noted especially in CD4-depleted animals (Fig. 6a). In stark contrast, depletion of both T-cell subsets in $Tlr^{-/-}$ mice converted tumour rejection into progression with tumour growth curves resembling TLR-competent WT control mice. These data clearly establish that tumour rejection in the absence of TLR3/7/9 depends on the activity of both CD4 and CD8 T cells. Myeloid cells instead do not alter tumour rejection, as $Tlr^{-/-}$ rejected tumours also during Gr-1 depletion in three out of four cases (Fig. 6b).

In the final part of our study, we asked whether the T-cell-dependent tumour rejection results in a classical memory response. To address this point, we performed a re-challenge experiment (Fig. 6c). $Tlr^{-/-}$ mice were injected with MOPC tumours and allowed to reject the tumour (blue squares in fig. 6c). After rejection, the same number of tumour cells was injected into the opposite flank of the regressor mice (arrow in fig. 6c). At this time point, naive WT and additional TLR-deficient mice were also injected with tumour cells (arrow) and served as controls. As expected, WT mice showed progressive tumour growth (black circles), whereas tumour-naive TLR-deficient mice (grey triangles) showed tumour rejection ~10

days after tumour cell injection. No tumour growth could be observed in TLR-deficient mice, which previously rejected the tumour (blue squares), demonstrating the induction of an anti-tumour memory response. Collectively, our experiments reveal an unexpected capacity of TLR3/7/9-deficient mice to induce protective T-cell-dependent immunity, which is associated with substantial changes in the tumour microenvironment, T-cell activation and tumour regression.

**Discussion**

TLRs are important sensors of pathogen-associated molecular patterns and help to protect the host from foreign intruders. Engagement of TLRs links innate with adaptive immunity and results in the induction of protective immunity. This principle has subsequently been used and exploited therapeutically, making TLR agonists important components in today's cancer immunotherapy. Conversely, and somewhat counterintuitive, the triggering of TLRs by endogenous ligands in some cases does not result in anti-tumour immunity but rather contributes to tumour progression. This can be explained by the fact that in these cases TLR activation is a continuous and chronic process, which is induced by either tumour-inducing pathogens[30,31] or by endogenous TLR ligands released during cancer-related inflammation[32]. More recently, the interconnection of the so-called microbiome with endogenous and therapy-induced anti-tumour immune responses has increasingly gained attention[33]. In a pioneering paper using murine melanoma models, Sivan et al.[34] identified distinct commensal microbiota, which were associated with improved endogenous anti-tumour immunity and synergized with checkpoint-blockade immunotherapy. In a model of therapy-induced anti-tumour immunity, the interaction of commensal microbiota and myeloid cells influenced response to immuno- and chemotherapy[35]. All in all, these data highlight the complex and highly context-dependent interconnection of TLR signalling and anti-tumour immunity.

Both tumour cells and non-malignant cells of the tumour host can be targets for TLR ligands. It has been described that triggering of TLR responses in tumour cells can lead to the induction of tumour cell death[7], which, under certain conditions, is associated with induction of anti-tumour immunity by mechanisms referred to as immunogenic or immunostimulatory cell death[10]. Alternatively, activation of TLR on tumour cells can also promote tumour growth by inducing anti-apoptotic mechanisms, immune escape or chemoresistance[8,36].

In this study, we specifically focused on the role of the endosomal TLRs 3, 7 and 9 expressed on non-malignant host cells. To this end, we used transplantable WT tumour cell lines and TLR-deficient host mice. We show that the absence of those TLRs on host cells results in the induction of protective CD8 T-cell-mediated immunity and complete regression of WT tumour cells in multiple tumour models. Tumour regression required the cooperative activity of immune and radioresistant host cells. The anti-tumour immune response was associated with significant alterations in the tumour microenvironment such as vessel normalization, enhanced numbers and strong activation of tumour-infiltrating lymphocytes.

Until today, endogenous triggering of TLRs on host cells has been studied mainly for TLR2 and 4, whereas less is known on the endosomal TLRs 3, 7 and 9. This is partly due to the fact that the endogenous ligands, commonly referred to as DAMPs, for TLR2/4 are by far better characterized than their endosomal nucleic-acid-sensing counterparts. Examples for TLR2/4 DAMPs include HMGB1, histones, syndecans, heat-shock proteins and S100 proteins[37]. Interestingly, effects of TLR2 and TLR4 activation by those endogenous DAMPs seem to be context-dependent

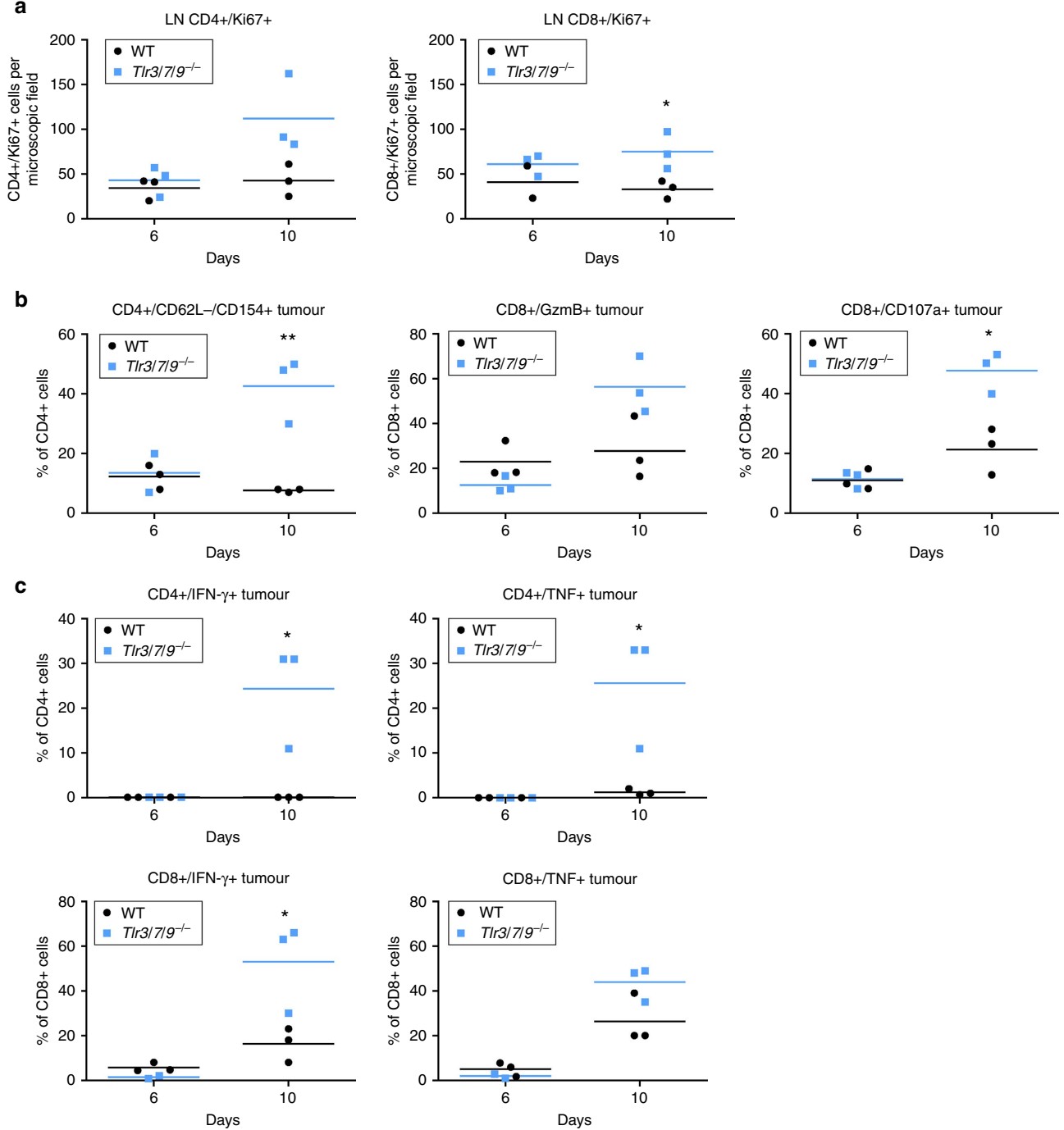

**Figure 5 | T-cell activation in lymph nodes and tumour. (a)** Lymph nodes: number of proliferating cells within the tumour-draining lymph nodes (LNs) of 6- to 12-week-old C57BL/6 WT and $Tlr3/7/9^{-/-}$ mice was assessed at day 6 and day 10 by immunofluorescence double staining of frozen sections using αCD4 and αKi67 antibodies. Quantification of positive cells was performed by observer-assisted counting on four randomly selected × 200 magnified images per sample ($n = 3$). **(b,c)** Tumour: flow cytometric analysis of T-cell activation of tumour-infiltrating lymphocytes of 6- to 12-week-old C57BL/6 WT and $Tlr3/7/9^{-/-}$ mice at day 6 and day 10. **(b)** Activation status of $CD4^+$ cells was analysed by staining of $CD62L^-/CD154^+$ cells. Activated $CD8^+$ cells are determined by showing Granzyme $B^+$ (GzmB) and $CD107a^+$ expression. **(c)** Cytokine production of tumour-infiltrating $CD4^+$ and $CD8^+$ cells was assessed after ex vivo stimulation with αCD3/CD28 antibodies and subsequent flow cytometry staining using αIFN-γ and αTNF-α antibodies. Single values and mean (horizontal line) are shown. Results were considered significant at *$P \leq 0.05$ and ** $P \leq 0.005$ (t-test).

as well. In a transplantable model of orthotopic head and neck cancer in C3H mice, the absence of TLR4 on host cells resulted in accelerated tumour growth[28]. In the context of chemo- or radiotherapy, the expression of TLR4 and activation of MyD88 mainly on dendritic cells were crucial for efficient induction of anti-tumour immunity[20]. Conversely, in models of melanoma

and bladder cancer, TLR4 signalling, supposedly via tumour-derived heat-shock proteins and tumour-associated macrophages, accelerated the growth of lung metastases[38]. Similarly, tumour-derived versican has been proposed as a driver of metastasis by promoting the pro-metastatic activity of myeloid cells via TLR2 (ref. 39). In contrast, the activation of TLR2 by HMGB1 during

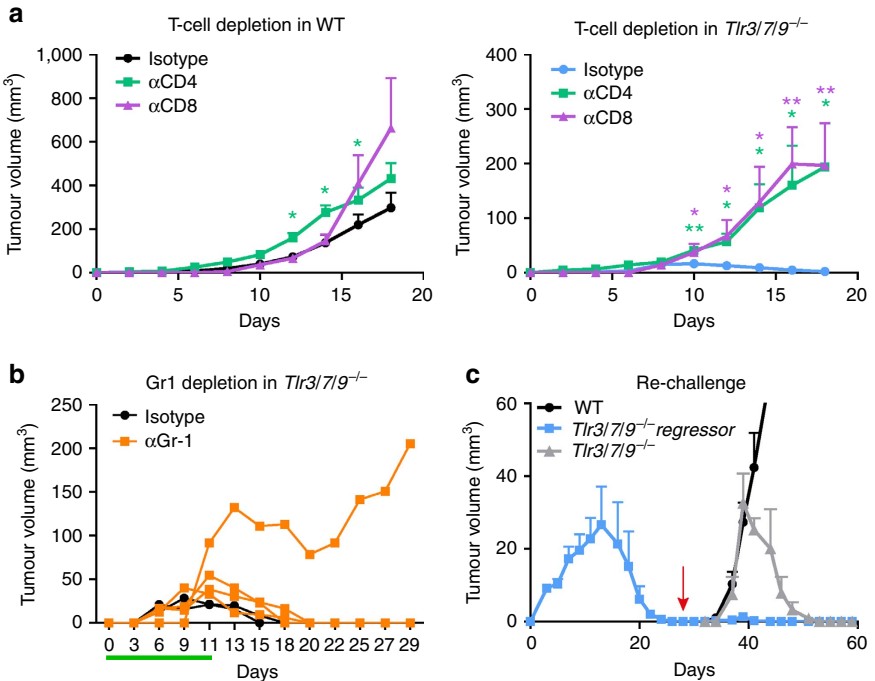

**Figure 6 | T-cell-dependent tumour rejection in _Tlr3/7/9_ $^{-/-}$ mice. (a)** Depleting antibodies against CD4 and CD8 were injected into 12- to 17-week-old C57BL/6 WT and _Tlr3/7/9_ $^{-/-}$ mice. Control mice were treated with appropriate isotype. Tumour growth was measured every other day. Mean ± s.e.m. of four to eight mice per group. Results were considered significant at *$P \le 0.05$ and **$P \le 0.005$ ($t$-test). **(b)** To deplete myeloid regulatory cells, treatment of 11- to 17-week-old C57BL/6 _Tlr3/7/9_ $^{-/-}$ mice with the αGr-1 antibody started before tumour cell injection and was continued throughout the whole experiment ($n = 4$). The control group was treated with isotype antibodies ($n = 2$). Depletion was monitored in the peripheral blood using flow cytometry and is indicated by the green line. One representative experiment out of two is shown. **(c)** Tumour cells ($0.5 \times 10^6$) were injected s.c. into the right flank of _Tlr3/7/9_ $^{-/-}$ mice (_Tlr3/7/9_ $^{-/-}$ regressor, blue). After tumour rejection (red arrow), the same mice were injected on the opposite flank with the same number of tumour cells. Naive _Tlr3/7/9_ $^{-/-}$ (grey) and C57BL/6 WT mice (black) were also injected and served as controls. At the time of rechallenge, mice were 24–26 weeks old. Tumour size was measured every other day. The mean tumour volume ± s.e.m. of three to five mice per group is shown.

brain tumour therapy was reported to contribute to anti-tumour immunity via activation of dendritic cells[21].

Unfortunately, and in contrast to TLR2/4, the endogenous ligands for TLR3/7/9 remain largely enigmatic. Some studies suggest that (mitochondrial) DNA, mostly in the context of severe injury and sepsis, could be implicated in pathological activation primarily of TLR9 (ref. 37). In addition, HMGB1 has been suggested as a DAMP for TLR9 (ref. 40), although it is difficult to determine whether HMGB1 or associated DNA is the recognized molecular structure[40,41]. In most of the studies mentioned above, single TLR-KO, mostly for TLR2 and TLR4, have been used and some studies were indeed able to identify a major DAMP, which modulates tumour growth in the models tested. There are two fundamental differences between those studies and ours. At first, in our study single and even double KO mice did not show tumour regression. Regression was only observed in the combined TLR3/7/9 triple KO. This could point to a complex interaction of potential TLR ligands or the simultaneous activity of multiple ligands in our system. Alternatively, compensatory mechanisms of nucleic-acid-sensing TLRs could be in place, which mask the potential phenotype of single TLR-KO mice. Second, and even more importantly, we demonstrate not only quantitative modulation of tumour growth/progression but rather complete regression in the absence of TLR3, 7 and 9. To the best of our knowledge, such an effect on tumour growth in the absence of TLRs has not been demonstrated previously. Thus, our data suggest that a comprehensive deficiency in the endosomal nucleic-acid-sensing system of the host is required and sufficient to induce complete T-cell-mediated tumour regression and induction of protective memory.

We tried to determine whether TLR deficiency on haematopoietic or non-haematopoietic radioresistant host cells was required for induction of tumour regression. Unexpectedly, only the combined deficiency of both compartments resulted in induction of anti-tumour immunity. This was somewhat surprising, as previous papers suggested a predominance of the haematopoietic cells in TLR-dependent modulation of tumour growth. In a key study demonstrating the TLR-dependent pro-tumorigenic activity of haematopoietic cells of the host, Kim _et al._[39] identified the inflammatory versican-TLR2-macrophage axis as a driver of metastasis. In a model of wound-induced skin cancer, the absence of TLR5 on leukocytes reduced tumour prevalence in BM chimeras[42]. Similarly, the absence of TLR7 on BM-derived cells reduced carcinogenesis in a pancreatic cancer model[43]. Conversely, implanted prostate cancer cells grew faster in _Tlr3_ $^{-/-}$ mice, although the relative contribution of BM-derived versus radioresistant cells or immune versus non-immune cells was not assessed in that study[44].

In our study, tumours in WT mice were mostly non-inflamed, with low numbers of tumour-infiltrating T cells, which showed little activation. In TLR-deficient tumour regressor mice, we observed clear signs of T-cell activation in both lymph nodes and tumours. This activation was accompanied by strongly enhanced numbers of T cells infiltrating tumours from TLR-deficient mice. In addition to T-cell activation itself, T-cell recruitment to the tumour tissue and anti-tumour immunity may also be influenced by vessels and stromal fibroblasts. Although the role of the so-called cancer-associated fibroblasts in tumour progression and modulation of tumour immunity is fairly well established[45,46], the influence of endothelial cells and especially pericytes on effector

cell recruitment to the tumour is only beginning to emerge[47]. In an interesting study by Johansson et al.[48] TNF-α and IFN-γ were targeted to the tumour vasculature. Although IFN-γ primarily had anti-vascular pro-apoptotic activity, low-dose TNF-α stabilized the tumour-associated vascular network and enhanced CD8 effector T-cell activity. During the phase of tumour regression, we observed altered vessel structure and increased pericyte coverage, which could indicate a potential role of the altered vasculature in enhanced recruitment of effector T cells into the tumour microenvironment. However, initial experiments performed in this model did not suggest induced vessel activation or altered vascular permeability and leakiness in TLR-deficient mice. Thus, at this stage it remains unclear whether vascular alterations and enhanced T-cell influx are mechanistically linked or represent rather independent parallel events. It is also important to note that changes in vessel structure, T-cell influx, T-cell activation and tumour regression seem to be rather parallel events in this model. All of these were observed around day 10, but were not yet present at day 6 after tumour injection.

In clinical studies and human cancer patients, mostly expression of TLR on tumour cells has been analysed and evaluated for prognostic impact and clinical relevance. Few studies investigated TLR expression on stromal cells. For example, in an observational clinical study, TLR9 on stromal fibroblasts predicted enhanced survival in breast cancer[16]. Similarly, high expression of TLR7 on stromal fibroblasts was associated with good prognosis in oral squamous cell carcinoma (OSCC)[49]. Conversely, expression of TLR4 on stromal fibroblasts was associated with poor prognosis in colorectal cancer[14]. Similar data were reported for hepatocellular carcinoma, where TLR9 expression on tumour-associated fibroblasts was associated with reduced overall survival[50]. Thus, our study is one of the first to experimentally demonstrate a role for radioresistant (stromal) cells in regulation of tumour progression via endosomal TLR signalling.

Collectively, our data uncover a novel function of endosomal nucleic-acid-sensing TLRs for the regulation of anti-tumour immune responses. The combined absence of three TLRs on both BM-derived and radioresistant cells of the tumour host was required for tumour regression. Our data also suggest a previously unknown role for endosomal TLRs in limiting endogenous adaptive and protective T-cell responses in the tumour host.

## Methods

**Animals.** All animal experiments were approved by the animal ethics committee of the state of North Rhine-Westphalia and carried out according to German guidelines for experimental animal welfare. Six- to 17-week-old male and female C57BL/6J WT and $Tlr3/7/9^{-/-}$ were obtained from the animal facility of the University Hospital Essen or from Harlan Winkelmann GmbH (Borchen, Germany) and housed under standard conditions in individually ventilated cage racks. Six- to 12-week-old male and female Tlr single and Tlr double KO mice were provided by Stefan Bauer, Philipps-University Marburg.

**Tumour cell culture.** The murine oropharyngeal cell lines (MOPCs) were kindly provided by J. Lee (Sanford Research/University of South Dakota) and cultured in medium containing DMEM high glucose and Ham's F12 (3:1) (Thermo Fisher Scientific, Karlsruhe, Germany) supplemented with 10 % (v/v) heat-inactivated FCS (Biochrom, Berlin, Germany), 100 IU ml$^{-1}$ penicillin, 100 µg ml$^{-1}$ streptomycin (Thermo Fisher Scientific), epidermal growth factor (5 µg ml$^{-1}$) (Biochrom), insulin (5 µg ml$^{-1}$), transferrin (5 µg ml$^{-1}$), cholera toxin (0.0084 µg ml$^{-1}$), hydrocortisone (0.5 µg ml$^{-1}$) and tri-iodo-thyronine (0.00136 µg ml$^{-1}$) (Sigma-Aldrich, Taufkirchen, Germany)[51]. MOPC$^{-eGFP}$ cells were generated by lentiviral gene transfer using a pCL6IEGwo empty vector[52] and enriched to >90 % enhanced green fluorescent protein (eGFP) positivity by FACS analysis. The murine lung carcinoma cell line TC1 was kindly provided by Z. Fridlender (Hadassah Medical Center, Israel) and the murine bladder cancer cell line MB49 by K. Esuvaranathan (University of Singapore, Singapore). Both cell lines were cultured in DMEM high glucose supplemented with 10 % (v/v) FCS, 100 IU ml$^{-1}$ penicillin, 100 µg ml$^{-1}$ streptomycin and 1 mM sodium pyruvate

(Thermo Fisher Scientific). All cells were routinely tested for mycoplasma contamination using the Venor Mycoplasma Detection Kit (Minerva Biolabs, Berlin, Germany).

**Tumour models.** Syngeneic murine cell lines were either injected s.c. into the flank or into the Musculus mylohyoideus (orthotopic injection). Tumour cells ($0.5 \times 10^6$–$2 \times 10^6$) were applied in PBS using a 27-gauge syringe under anaesthesia. After s.c. injection, tumour volume was measured every other day in two dimensions and tumour volume was estimated according to the following formula

$$v(\text{volume}) = \left(\frac{\pi}{6}\right) \times w(\text{width})^2 \times l(\text{length})$$

After orthotopic injection, weight of the mice was measured daily. Mice were killed after a weight loss of >20 % in accordance with animal welfare guidelines.

**Isolation of primary spleen cells and BM cells.** For splenocyte and lymph node cell isolation, tissue was dissociated with a plunger. BM was collected from femurs by flushing the bone with a 23-gauge needle. Erythrocyte lysis in splenocyte and BM was performed by applying distilled water to the cell pellet for 20 s before readjusting osmolarity with $2 \times$ PBS. Cell suspensions were passed through a 50 µm filter and cultured in RPMI-1640 (Thermo Fisher Scientific) supplemented with 10 % (v/v) heat-inactivated FCS, 100 IU ml$^{-1}$ penicillin and 100 µg ml$^{-1}$ streptomycin or in PBS.

T cells were isolated with the PanT cell isolation Kit (Miltenyi, Bergisch Gladbach, Germany) according to the manufacturer's instructions. Purity was >90 % in all experiments.

**Isolation of cells from tumour tissue.** Tumour tissue was cut into 2–3 mm pieces in PBS, centrifuged at $300\,g$ for 7 min and resuspended with 1 ml enzyme mix containing 0.2 mg ml$^{-1}$ Collagenase D, 0.2 mg ml$^{-1}$ Dispase 1 and 0.1 mg ml$^{-1}$ DNase (all from Sigma-Aldrich). After 45 min incubation at 37 °C, the cells were passed through a 100 µm filter and kept in RPMI-1640, supplemented as described above, until analysis.

**Generation of BM chimeric mice.** C57BL/6 WT and $Tlr3/7/9^{-/-}$ mice were exposed to one dose (9.5–10.5 grey) total body irradiation, from a γ-beam $^{60}$Co source (Philips, Germany). Donor BM cells from C57BL/6 WT and $Tlr3/7/9^{-/-}$ mice were isolated as described above. Recipient mice were injected intravenously with $5 \times 10^6$ BM cells in 0.2 ml PBS 1 day after irradiation. BM from WT mice was injected into TLR-deficient mice and vice versa. Mice receiving BM of the same genotype served as control. To achieve haematopoietic reconstitution, mice were allowed to recover for 8 weeks until $0.5 \times 10^6$ MOPC tumour cells were injected s.c. into the flank of the mice. Tumour growth was measured every other day for 1 month.

**Antibody-mediated depletion.** αGr-1 antibody (RB6-8C5, BioXCell, West Lebanon, USA) was used to deplete Gr-1-positive cells by intraperitoneal injection of 200 µg antibody diluted in 0.9 % NaCl. Treatment started 3 days before tumour cell injection and was repeated every third day throughout the experiment. For depletion of CD8 or CD4 T cells, 100 µg antibody (CD8, 2.43, BioXCell and CD4, GK1.5, BioXCell) was used the same way with application intervals of 7 days. Control groups were treated with isotype antibody (rat IgG2a LTF-2, BioXCell) using the same regimen as applied for the specific antibody. Depletion was verified using peripheral blood from tail veins and flow cytometry.

**Cytokine analysis of cell culture SNs.** Explanted tumour pieces with a mean weight of 12 mg were cultured at 20 mg ml$^{-1}$ in medium for 48 h to generate tumour explant SNs. SNs were collected, centrifuged to remove cell debris and stored at $-20$ °C until analysis.

The murine cytokines IL-6 and IL-24 were detected using DuoSet ELISA Kits (R&D Systems, Wiesbaden, Germany) and mouse IFN-γ was detected using OptEIA Set ELISA Kit (BD Biosciences, Heidelberg, Germany) according to the manufacturer's protocol. Absorbance at 450 nm was measured with a Synergy II microplate reader (BioTek, Bad Friedrichshall, Germany).

**Flow cytometry.** Cells were stained with fluorochrome-coupled antibodies in PBS supplemented with 5% murine serum and 10% FCS for 20 min at 4 °C. The following antibody conjugates and respective clones were used: CD11b-PE-Cy7 (clone 3A33, 2 µg ml$^{-1}$, Abcam, Cambridge, UK), Ly6G-PerCP-Cy5.5 (clone 1A8, 2 µg ml$^{-1}$, BioLegend, Fell, Germany), CD4-V450 (clone 2B11, 4 µg ml$^{-1}$, BD Biosciences), CD4 (BV605, RM4-5, 0.25 µg ml$^{-1}$, BioLegend), CD8 (AF700, 53-6.7, 1 µg ml$^{-1}$, eBioscience), CD8-V500 (clone 53-6.7, 4 µg ml$^{-1}$, BD Biosciences), CD3 (AF700, 17A2, 1 µg ml$^{-1}$, eBioscience), CD107a (fluorescein isothiocyanate, 1D4B, 0.5 µg ml$^{-1}$, BioLegend), CD43 (PerCP, 1B11, 0.25 µg ml$^{-1}$, BioLegend), CD62L (BV510, Mel14, 0.25 µg ml$^{-1}$ BioLegend), CD154 (PE, MR1, 0.25 µg ml$^{-1}$, eBioscience), CD244.2 (APC, eBio244F4, 1 µg ml$^{-1}$, eBioscience), LAG3 (PE, eBioC9B7W, 0.25 µg ml$^{-1}$, eBioscience),

PD-1 (PE, J43, 1 µg ml$^{-1}$, eBioscience) and Tim-3 (PerCP, 215008, 1:100, R&D). Dead cells (Fixable Viability Dye eFluor 780 positive, eBioscience) and doublets were excluded from analyses. Intracellular staining of granzyme B (APC, GB12, 0.25 µg ml$^{-1}$ Life Technologies), Eomes (PerCp-eF710, Dan11mag, 1 µg ml$^{-1}$, eBioscience) and FoxP3 (PE, eFlour610 FJK-16s, 1 µg ml$^{-1}$, eBioscience) was performed after fixation and permeabilization step with Foxp3 Fixation kit (Foxp3/Transcription Factor Fixation/Permeabilization Concentrate and Diluent, eBioscience) according to the manufacturer's protocol.

For intracellular cytokine staining, the lymph node cells and tumour cells were incubated with plate-bound anti-CD3 (145-2C11, 4 µg ml$^{-1}$, BioLegend) and soluble anti-CD28 (37.51, 2 µg ml$^{-1}$, BioLegend) for 4 h at 37 °C in the presence of brefeldin A (2 µg ml$^{-1}$, Sigma-Aldrich). Cells were washed twice, incubated with anti-Fcγ 2/3 receptor (2.4G2, 1 µg ml$^{-1}$, eBioscienc) to block Fc receptors and stained for surface markers. The cells were then washed and permeabilized using the Cytofix/Cytoperm intracellular staining kit (Becton Dickinson) and reacted with monoclonal antibodies specific for IL-2 (eF450, JES6-5H4, 0.5 µg ml$^{-1}$, eBioscience), IFN-γ (fluorescein isothiocyanate, XMG1.2, 0.5 µg ml$^{-1}$, eBioscience) and TNF-α (BV510, MP6-XT22, 1 µg ml$^{-1}$, BioLegend). Isotype control antibodies were used at the same concentration as the specific antibodies.

Data were recorded at a FACS CantoII using DIVA 6.0 software (BD Biosciences) or on a LSR II (BD Biosciences) using FlowJo (Treestar).

**Gene expression microarray analysis.** MOPC tumour cells ($2 \times 10^6$) were injected s.c. into C57BL/6 WT and $Tlr3/7/9^{-/-}$ mice. After 6 days, tumours were dissected and immediately frozen in liquid nitrogen. Tumour tissue was mechanically minced and total RNA was isolated following the instructions of the RNeasy MINI Kit (Qiagen, Hilden, Germany). RNA concentration and purity were controlled using the micro cuvette G1.0 (Eppendorf, Wesseling-Berzdorf, Germany) and a BioPhotometer (Eppendorf). Two hundred and fifty nanograms of RNA was transcribed into complementary DNA using the 3′-IVT Expression Plus Kit (Affymetrix, High Wycombe, UK). cDNA was analysed using the Affymetrix Chip (MG-430_2.0) in the Biochip core facility of the Medical Faculty of the University Duisburg-Essen.

**Gene expression analysis by quantitative RT–PCR.** For quantitative RT–PCR analysis, total RNA was isolated from tumour tissue using the RNeasy kit (Qiagen) and reverse transcribed with random-hexamer primer and Superscript II RT, according to the manufacturer's instructions (Thermo Fisher Scientific). Quantitative real-time PCR was conducted with Maxima SYBR Green quantitative PCR Master Mix (Thermo Fisher Scientific, Bonn, Germany)[53]. Primers are listed in Supplementary Table 1. Annealing temperature was 60 °C for all primers.

**Histology and immunohistochemistry of tumour tissue.** Tumour tissue was frozen in optimal cutting temperature (OCT) embedding medium on liquid nitrogen and stored at − 80 °C. Frozen sectioning was performed at 5 µm thickness. For morphological analysis, tumours were stained by haematoxylin–eosin. Colorimetric immunohistochemistry was performed after fixation of tissue with the Cytofix/Cytoperm Kit (BD Biosciences) and incubation with hydrogen peroxide (DAKO, Hamburg, Germany). Slices were incubated for 60 min at room temperature with the primary monoclonal antibody in a humid chamber. Subsequently, samples were incubated with secondary antibodies for 30 min and for 10 min with AEC Single Solution (Thermo Fisher Scientific). Nuclei were visualized by Shandon Instant Hematoxylin (Thermo Fisher Scientific)[28]. The following primary (rat-α-mouse) and secondary antibodies were used: rat αCD31 (clone MEC13.3, 1:1,000), rat αGR1 (clone RB6-8C5, 1:100), rat αCD4 (clone RM4-5, 1:500, all from BD Biosciences), rat αCD11b (clone M1/70.15, 1:1,000; Thermo Scientific), rat αCD8α (clone KT15, 1:500, AbD Serotec/Bio-Rad, Düsseldorf, Germany), rabbit αDesmin (ab15200, 1:600, Abcam), rabbit α-rat HRPO, goat α-rabbit HRPO (both 1:100, Dianova, Hamburg, Germany), donkey α-rabbit-Alexa594 (1:800) and goat α-rat-Alexa488 (1:200) (both from Thermo Fisher Scientific).

Double fluorescence staining of tumour tissue was performed after fixation with BD Cytofix/Cytoperm overnight at 4 °C with rat αCD31 (1:1,000) and rabbit αCaspase 3 (active) (1:100, R&D Systems), and rat αCD4 (1:500) or rat αCD8 (1:500) together with sheep αKi67/MKI67 (1:200, R&D Systems), followed by staining for 30 min at room temperature with the secondary reagents goat-α-rabbit Cy3, donkey-α-sheep Cy3 (both 1:400, Dianova) and goat-α-rat Alexa488 (1:200, Thermo Scientific). Nuclei were visualized by 4,6-diamidino-2-phenylindole staining (1:36,000, BioLegend). For verification of vascular maturation, desmin-positive vessels were quantified by manual counting of up to five fields per tumour, including 100 vessels in total, using the mean of three tumour sections per tumour (top, middle and base) of concordant distance[29].

For quantification of tumour-infiltrating lymphocytes and vessels, images were taken using Axioskop 2 (Carl Zeiss, Jena, Germany) and AxioVision (Carl Zeiss) software as indicated in the experiment. Cells were quantified by manual counting of individual cells of four microscopic fields taken at × 200 magnification. Quantification was done by independent investigators in a blinded manner. Quantification of pericyte coverage was performed by manual counting of up to five fields per tumour, including 100 vessels in total, using the mean of three tumour section per tumour (top, middle and base) of concordant distance. Vessel

analysis was performed by using the BX51 (Olympus, Hamburg, Germany) and corresponding Cell P Software. For representative pictures, images were processed in the same file simultaneously in Photoshop CS5 (Adobe Systems, San Jose, USA).

**Determination of cell death.** Apoptosis of cultured cells was detected using AnnexinV:PE Apoptosis Detection Kit (BD Biosciences) via flow cytometry, according to the manufacturer's protocol. In frozen sections of tumour tissue, total amounts of apoptotic cells were quantified by APO-DIRECT TUNEL staining (BD Biosciences). 7-Aminoactinomycin D (7-AAD) (BD Biosciences) was used as nuclear staining. To determine the number of TUNEL-positive cells in relation to the whole tumour area, digital images taken at × 25 magnification were analysed using Image J software (NIH, Bethesda, USA).

**Tumour clearing and ultramicroscopy.** MOPC$^{-eGFP}$ cells ($1 \times 10^6$) were injected s.c. into WT and $Tlr3/7/9^{-/-}$ mice. If vessel staining was required, 10 µg of the rat-α-mouse CD31 (clone MEC13.3, BioLegend) antibody coupled with Alexa Fluor 647 was injected retrobulbarly in each mouse (20 µg ml$^{-1}$). Injection was carried out under anaesthesia by intraperitoneal application of ketamine and xylazine dose at 100 and 10 mg per kg body weight, respectively, and diluted in 0.9 % NaCl for injection of 10 µl per g body weight. Twenty minutes later, mice were killed and perfused by injection of 20 ml 5 mM EDTA/PBS (pH 7.2) into the left ventricle of the heart followed by 20 ml 4 % paraformaldehyde (PFA). Tumours were dissected and the 3DISCO protocol was used for clearing[54]. Briefly, tumours were incubated in tetrahydrofuran with increasing concentration (30, 50, 80, 100% (v/v)), for 30 min each. Subsequent incubation in dibenzyl ether was performed overnight. Vascularization of the tumours was analysed using the lightsheet illumination technique of the LaVision BioTec Ultramicroscope (Olympus). Data were analysed using the ImageJ and Imaris 8.0.2 Software (Imaris, Cologne, Germany).

**Adoptive transfer.** Recipient WT and $Tlr3/7/9^{-/-}$ mice were s.c injected with $1 \times 10^6$ MOPC$^{-eGFP}$ tumour cells 6 days before adoptive transfer. Donor mice were s.c. injected with $1 \times 10^6$ MOPC$^{-eGFP}$ tumour cells 10 days before harvesting of donor splenocytes. Splenocytes of WT and $Tlr3/7/9^{-/-}$ donor mice were isolated and labelled with 1 µM CellTracker Deep Red Dye (Thermo Scientific). Labelled splenocytes ($10 \times 10^6$) were injected intravenously into recipient mice. One day later, tumours were dissected and cleared. Infiltration of transferred splenocytes was examined by generating 3D images using ultramicroscopy.

**Statistical analysis.** Statistical significance was assessed with the Student's $t$-test for comparison of two groups or analysis of variance if more than two groups were compared. Number of mice per group is indicated for each experiment and individual mice are shown for most experiments. Sample size was determined based on previous preliminary data. No animals were excluded from analysis and no explicit method of randomization was used to assign animals to experimental groups. Results were considered significant at $*P \leq 0.05$, $**P \leq 0.005$ and $***P \leq 0.0005$. Hierarchical clustering of gene expression data was performed with R software (The R Foundation for Statistical Computing, Vienna, Austria).

**Data availability.** Microarray data supporting the findings in this article were deposited in NCBI's Gene Expression Omnibus under the GEO Series accession code GSE92358. All other data generated or analysed during this study are included in this published article and its Supplementary Information files or available from the corresponding author upon reasonable request.

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

## Acknowledgements

We thank Kirsten Bruderek, Sebastian Vollmer and Petra Altenhoff for their excellent technical assistance, Stefan Bauer (Philipps-University, Marburg) for providing the single and double $Tlr^{-/-}$ mice and John Lee (Sanford Research, University of South Dakota) for providing the MOPC cell line. We especially thank Ali Sak and Michael Groneberg (Department of Radiotherapy, University Hospital Essen) for assistance with generation of BM chimera mice, and Bernd Giebel and Andre Görgens (Institute for Transfusion Medicine, University Hospital Essen) for help with generation of the fluorescent MOPC$^{-eGFP}$ cell line. We thank Ludger Klein-Hitpaß (Biochip lab of University Hospital Essen) for performing whole-mouse genome gene expression analysis and for valuable help with the gene expression analysis. We thank the staff of the IMCES Imaging Center Essen for expert technical support and Marc Seifert for help with submission of array data.

## Author contributions

J.C.K. planned and performed experiments, analysed and interpreted data, and wrote the manuscript. K.M. and G.Z. performed experiments. I.H. performed experiments, analysed data and wrote parts of the manuscript. S.S. analysed data. J.B. and S.L. provided financial and infrastructural support. U.D. designed experiments. C.J.K. provided financial support and most KO mice and contributed to experimental planning and data interpretation. S.B. provided financial support and conceived of the study, interpreted data and wrote the manuscript.

## Additional information

**Competing financial interests:** The authors declare no competing financial interests.

**Publisher's note**: 

